# Peptide Stapling Improves the Sustainability of a Peptide-Based Chimeric Molecule That Induces Targeted Protein Degradation

**DOI:** 10.3390/ijms22168772

**Published:** 2021-08-16

**Authors:** Hidetomo Yokoo, Nobumichi Ohoka, Mami Takyo, Takahito Ito, Keisuke Tsuchiya, Takashi Kurohara, Kiyoshi Fukuhara, Takao Inoue, Mikihiko Naito, Yosuke Demizu

**Affiliations:** 1Division of Organic Chemistry, National Institute of Health Sciences, 3-25-26, Kanagawa 210-9501, Japan; yokoo@koto.kpu-m.ac.jp (H.Y.); w205426e@yokohama-cu.ac.jp (M.T.); titou@nihs.go.jp (T.I.); p19940324@pharm.showa-u.ac.jp (K.T.); tks-kurohara@nihs.go.jp (T.K.); 2Medical Chemistry, Graduate School of Medical Science, Kyoto Prefectural University of Medicine, Kyoto 606-0823, Japan; 3Division of Molecular Target and Gene Therapy Products, National Institute of Health Sciences, 3-25-26, Kanagawa 210-9501, Japan; n-ohoka@nihs.go.jp (N.O.); takao@nihs.go.jp (T.I.); 4Graduate School of Medical Life Science, Yokohama City University, Yokohama 230-0045, Japan; 5Graduate School of Pharmacy, Showa University, Tokyo 142-0064, Japan; fukuhara@pharm.showa-u.ac.jp; 6Laboratory of Targeted Protein Degradation, Graduate School of Pharmaceutical Sciences, The University of Tokyo, Tokyo 113-0033, Japan; miki-naito@g.ecc.u-tokyo.ac.jp

**Keywords:** estrogen receptors, helical peptide, protein-protein interaction, protein knockdown

## Abstract

Peptide-based target protein degradation inducers called PROTACs/SNIPERs have low cell penetrability and poor intracellular stability as drawbacks. These shortcomings can be overcome by easily modifying these peptides by conjugation with cell penetrating peptides and side-chain stapling. In this study, we succeeded in developing the stapled peptide stPERML-R7, which is based on the estrogen receptor alpha (ERα)-binding peptide PERML and composed of natural amino acids. stPERML-R7, which includes a hepta-arginine motif and a hydrocarbon stapling moiety, showed increased α-helicity and similar binding affinity toward ERα when compared with those of the parent peptide PERML. Furthermore, we used stPERML-R7 to develop a peptide-based degrader LCL-stPERML-R7 targeting ERα by conjugating stPERML-R7 with a small molecule LCL161 (LCL) that recruits the E3 ligase IAPs to induce proteasomal degradation via ubiquitylation. The chimeric peptide LCL-stPERML-R7 induced sustained degradation of ERα and potently inhibited ERα-mediated transcription more effectively than the unstapled chimera LCL-PERML-R7. These results suggest that a stapled structure is effective in maintaining the intracellular activity of peptide-based degraders.

## 1. Introduction

A variety of small molecule inhibitors that target intracellular enzymes have been developed as therapeutic agents. However, many pathogenic proteins do not have enzymatic activity, making them difficult to target with conventional inhibitors and requiring different approaches. Targeted protein degradation has attracted attention as an emerging approach in drug discovery [1,2]. Many target protein degraders, proteolysis targeting chimeras (PROTACs) and specific and non-genetic inhibitor of apoptosis protein [IAP]-dependent protein erasers (SNIPERs), have been developed using chimeric molecules (X-L) that consist of a target protein ligand (X) and an E3 ligase ligand (L) (Figure 1a) [3,4,5]. These chimeric molecules crosslink target proteins with E3 ligases to induce polyubiquitylation and subsequent proteasomal degradation of the target proteins in cells.

Small molecules are often used as target protein ligands and E3 ligands in the development of PROTACs and SNIPERs. For example, to target the estrogen receptor (ER) and androgen receptor (AR), small molecules, 4-hydroxytamoxifen for ER and enzalutamide for AR, have been used in the design of PROTACs and SNIPERs [6,7]. Small-molecule E3 ligands VH032, pomalidomide and LCL161 are often used to recruit the E3 ligase complex, including von Hippel Lindau, cereblon [8,9,10] and IAPs, respectively [11,12,13,14,15]. In contrast, peptide-based ligands are capable of interacting with the surface of proteins, which is expected to expand the applicability of PROTACs and SNIPERs [16]. However, peptide-based degraders have not been widely explored because of their poor chemical stability and low cell membrane permeability.

Stabilizing the secondary structures of oligopeptides, particularly helical structures, is one approach that can overcome the abovementioned problems. To stabilize the helical structures of oligopeptides, non-proteinogenic amino acids and side-chain stapling are often incorporated into the sequences [17,18]. Some of these chemically modified peptides have been used as both E3 ligands and target ligands for chimeric molecules that degrade ERα [16,19,20]. We have previously developed a peptide-based ERα degrader (LCL-PERM3-R7) composed of a the peptidomimetic estrogen receptor modulators 3 (PERM3) that binds to ERα with high affinity, a cell-penetrating peptide (hepta-arginine, R7), and small-molecule-based E3 ligands (LCL161) [21,22]. However, the PERM3 peptide contains a disulfide bond in its sequence, which is cleaved by the reductive environment in the cell and therefore loses its activity, which remains an unresolved issue [21]. As a potential alternative, the hydrocarbon-type staple structure is quite stable in a cell environment, and thus, was expected to be useful for the development of peptide-based ERα degraders. In this study, we designed and synthesized a helix-stabilized peptide (stPERML) by introducing hydrocarbon stapling into the PERML peptide, which consists of natural amino acids that bind to ERα with high affinity (Figure 1b) [23,24]. The *i*/*i* + 4 stapling of stPERML-R7 was introduced at the appropriate position based on the co-crystal X-ray structure of the PERM peptide and ERα (Appendix A) [25]. Furthermore, chimeric degraders linking these peptides with an IAP ligand (LCL161) were also designed to compare their degradation activity against ERα at the cellular level. LCL161 was conjugated to the ε-position of the Lys(1) residue via an ethylene glycol-based linker. Thus, we designed chimeric molecules, LCL-PERML-R7 and LCL-stPERML-R7, and investigated the effects of the stapling structure of PERML on the degradation activity and its sustainability (Figure 2).

## 2. Results

The designated peptides were synthesized using a conventional Fmoc-based solid-phase method (Scheme 1). After construction of the PERML-R7 moiety, the relevant E3 ligase ligand was conjugated to the ε-amino group of the N-terminal Lys residue via an ethylene glycol-based linker. All six peptides PERML, PERML-R7, stPERML, stPERML-R7, LCL-PERML-R7 and LCL-stPERML-R7 were purified by reversed-phase high-performance liquid chromatography. The synthetic route of LCL-stPERML-R7 is shown in Scheme 1.

Circular dichroism (CD) spectra of PREML, PERML-R7 and stPERML-R7 were recorded to assess the preferred secondary structures of the peptides (Figure 3). PREML and PERML-R7 showed negative maxima at approximately 196 nm, indicating that they adopt random coil structures. On the other hand, stPERML-R7 showed negative maxima at approximately 208 and 222 nm, indicating that stPERML-R7 formed a helical structure. Thus, the introduction of side-chain stapling into the PERML sequence facilitated formation of a stable helical structure.

To investigate whether the introduction of the staple into the peptide sequence affects the binding to ERα, the ERα binding affinity (IC_50_) of PERML and stPERML was determined using a competitive fluorescence polarization assay (Table 1). The IC_50_ values of PERML (0.7 ± 0.7 μM) and stPERML (1.1 ± 0.4 μM) were comparable. Thus, the substitutions of amino acid residues at the 4th and 8th positions of PERML did not significantly change the binding affinity to the ERα, suggesting that these substitutions are appropriate for the introduction of the staple.

We have previously developed a peptide-based ERα degrader, LCL-PERM3-R7 [21,22]. First, to investigate whether LCL-PERML-R7 and LCL-stPERML-R7 have the degradation activities against ERα like LCL-PERM3-R7, the effect of the peptides on the protein level of ERα were evaluated by western blotting analysis using MCF-7 breast cancer cells. The ERα degradation activities of LCL-PERML-R7 and LCL-stPERML-R7 after 8 h of treatment were similar and slightly higher when compared with that of LCL-PERM3-R7 (Figure 4a). This result suggests that PERML and stPERML can be applied to the development of ERα protein degraders. To confirm the mechanism of ERα reduction by these peptides, we examined the effects of ubiquitin-proteasome system (UPS) inhibitors (Figure 4b). The peptides-mediated reduction in ERα was abrogated by co-treatment with a proteasome inhibitor, MG132, and a ubiquitin-activating inhibitor, MLN7243, indicating that the peptides induce UPS-dependent degradation of ERα.

Since the hydrocarbon-type staple structure is quite stable in the intracellular environment, the activity of peptide with this structure may be sustained. Next, to investigate whether the stapling of the peptides contributes to the sustainability of ERα degradation, the degradation activities of LCL-PERML-R7 and LCL-stPERML-R7 were compared at 8, 24, 48 and 72 h (Figure 4c). The degradation activity of LCL-PERML-R7 almost disappeared after 24 h, whereas that of LCL-stPERML-R7 remained until at least 72 h. This result showed that LCL-stPERML-R7 maintained degradation activity for a longer period than LCL-PERML-R7, and this sustained activity may be due to increased intracellular chemical stability of stPERML containing non-proteinogenic amino acids because the ERα binding affinity of stPERML was comparable to that of PERML (Table 1).

Finally, we investigated the effect of ERα degradation by these peptides on biological activity. The reporter gene assay with the peptides was performed to compare their inhibitory efficiency on 17β-estradiol (E2)-mediated ERα transcriptional activity (Figure 5). PERML and PERML-R7 showed no inhibitory activity up to a concentration of 20 µM. In contrast, LCL-PERML-R7 showed weak inhibitory activity at 20 µM and LCL-stPERML-R7 showed the most effective inhibitory activity at the same concentration, suggesting that higher degradation activity against ERα led the efficient functional inhibitory activity of ERα.

## 3. Discussion

We designed and synthesized the stapled peptide-based ERα degrader LCL-stPERML-R7 composed of stPERML, which binds to ERα, and LCL161 (LCL), which binds to an E3 ligase IAP. LCL-stPERML-R7 induced particularly high degradation activity for treatment periods longer than 24 h when compared with that of LCL-PERML-R7 containing the linear ERα-binding peptide PERML. The results indicate that stapling PERML is an effective approach to prolong the duration of degradation activity. In addition, the degrader LCL-stPERML-R7 showed higher transcriptional inhibition activity of ERα than that of the peptide ligand (stPERML) and degrader without the stapling structure (LCL-PERML-R7). The introduction of hydrocarbon stapling into the peptides could increase their chemical stability, which may be one of strategies to overcome limitations of peptide-based degraders, and also expand the availability of peptide ligands that can access the surfaces of proteins. The peptide-based degrader developed in this study is a multifunctional molecule: a protein-binding site, chemical modification for enhancing the secondary structure and chemical stability, a cell membrane-permeable fragment, and an E3 ligase ligand. We hope that these molecular modification strategies will lead to the development of highly functional PROTACs targeting a variety of proteins.

## 4. Materials and Methods

All of the coupling reagents were obtained from Tokyo Chemical Industry Co., Ltd. (Tokyo, Japan), and were used as supplied without further purification. Fmoc-protected amino acids were obtained from Tokyo Chemical Industry Co., Ltd. and Watanabe Chemical Industries, Ltd. (Hiroshima, Japan).

### 4.1. Synthesis and Characterization of the Peptides

NovaPEG Rink amide resin was initially soaked for 1 h in CH_2_Cl_2_/*N*,*N*-dimethylformamide (DMF). After the resin had been washed with DMF, the reactions with an Fmoc-amino acid were performed according method A or B.

Method A: Fmoc-protected amino acids (5 eq), diisopropylcarbodiimide (10 eq), Oxyma pure (10 eq) in DMF were used each coupling. The residues of Arg and (S)-pentenyl alanine (S5) were performed additional coupling using Fmoc-protected amino acids (6 eq), diisopropylcarbodiimide (6 eq), Oxyma pure (6 eq) in DMF. Each coupling was performed under microwave irradiation and temperature was 15 s at 75 °C, then 110 s at 90 °C. Deprotection of Fmoc group was performed using 20% piperidine in DMF under microwave irradiation at 75 °C for 15 s, then at 90 °C for 50 s.

Method B: Fmoc-protected amino acids (6 eq), hexafluorophosphate benzotriazole tetramethyl uronium (HBTU) (6 eq), 1-hydroxybenzotriazole (HOBt) (6 eq), *N*,*N*-diisopropylethylamine (DIPEA) (10 eq) in DMF were used each coupling. Each coupling was performed twice at 30 °C for 1 h. Deprotection of Fmoc group was performed using 40% piperidine in DMF at 30 °C for 3 min, then 20% piperidine in DMF at 30 °C for 12 min.

#### 4.1.1. Synthesis of PERML

After the peptide synthesis of PERML moiety (50 µmol scale) following Method A, the resin was suspended in cleavage cocktail [2 mL TFA, 50 μL water, 50 μL 1,2-ethanedithiol, 20 μL triisopropylsilane; final concentration: 94% TFA, 2.5% water, 2.5% 1,2-ethanedithiol, 1% triisopropylsilane] for 2 h at rt. The TFA solution was evaporated to a small volume under a stream of N_2_ and dripped into cold ether to precipitate the peptide.

#### 4.1.2. Synthesis of PERML-R7

Employing the procedure described above for PERML and starting from PERML-R7.

#### 4.1.3. Synthesis of stPERML

After the synthesis of peptide (50 μmol scale) following Method B without deprotecting N-terminus Fmoc group, ring-closing metathesis reactions were performed three times using 20 mol% second-generation Grubbs catalyst in DCE under microwave irradiation and temperature was at 70 °C for 12 min. Final deprotection was performed 4 mL of 20% PPD in DMF at rt for 10 min.

The resin was suspended in cleavage cocktail [2 mL TFA, 50 μL water, 50 μL triisopropylsilane; final concentration: 94% TFA, 2.5% water, 2.5% 1,2-ethanedithiol, 1% triisopropylsilane] for 2 h at rt. The TFA solution was evaporated to a small volume under a stream of N_2_ and dripped into cold ether to precipitate the peptide.

#### 4.1.4. Synthesis of stPERML-R7

After the synthesis of peptide (50 μmol scale) following Method A without deprotecting N-terminus Fmoc group, ring closing metathesis was performed using 20 mol% of second-generation Grubbs catalyst in DMF (2 mL) under N_2_ bubbling condition at room temperature overnight. Final deprotection was performed 2 mL of 20% piperidine DMF solution (*v*/*v*) for 20 min. The resin was suspended in cleavage cocktail [2 mL TFA, 50 μL water, 50 μL triisopropylsilane; final concentration: 94% TFA, 2.5% water, 2.5% 1,2-ethanedithiol, 1% triisopropylsilane] for 2 h at rt. The TFA solution was evaporated to a small volume under a stream of N_2_ and dripped into cold ether to precipitate the peptide.

#### 4.1.5. Synthesis of LCL-PERML-R7

After the peptide synthesis of PERML-R7 moiety (25 μmol scale) following method A, Fmoc-NH-PEG2-CO2H (4 eq), HBTU (4 eq), DIPEA (4 eq) and HOBt (4 eq) in DMF (2 mL) were added, and the reaction mixture was shaken for 1 hr. After deprotection of the Fmoc groups and repeating the above reaction, the N-Boc protected LCL161 (2 eq) was coupled with the peptide using HBTU (2 eq), DIPEA (4 eq), and HOBt (2 eq) in DMF (2 mL). The resin was suspended in cleavage cocktail [2 mL TFA, 50 μL water, 50 μL 1,2-ethanedithiol, 20 μL triisopropylsilane; final concentration: 94% TFA, 2.5% water, 2.5% 1,2-ethanedithiol, 1% triisopropylsilane] for 2 h at rt. The TFA solution was evaporated to a small volume under a stream of N_2_ and dripped into cold ether to precipitate the peptides.

#### 4.1.6. Synthesis of LCL-stPERML-R7

Employing the procedure described above for LCL-PERML-R7 and starting from stPERML-R7 afforded LCL-stPERML-R7.

#### 4.1.7. Synthesis of FAM-PERML

After the peptide synthesis of PERML moiety (25 μmol scale) following Method A, Fmoc-βAla-OH (4 eq), HBTU (4 eq), DIPEA (8 eq), HOBt (4 eq) in DMF (2 mL) were added, and the reaction mixture was shaken for 1 hr. After deprotection of the Fmoc groups, the 5,6-carboxyfluorescein (4 eq) was coupled with the peptide using HBTU (4 eq), DIPEA (8 eq), and HOBt (4 eq) in DMF (2 mL). The resin was suspended in cleavage cocktail [2 mL TFA, 50 μL water, 50 μL 1,2-ethanedithiol, 20 μL triisopropylsilane; final concentration: 94% TFA, 2.5% water, 2.5% 1,2-ethanedithiol, 1% triisopropylsilane] for 2 h at rt. The TFA solution was evaporated to a small volume under a stream of N_2_ and dripped into cold ether to precipitate the peptide.

The crude peptides were purified by reversed-phase high-performance liquid chromatography (HPLC) using a Discovery^®^ BIO Wide Pore C18 column (25 cm × 21.2 mm). Peptide purity was assessed using analytical HPLC and a Discovery^®^ BIO Wide Pore C18 column (25 cm × 4.6 mm), and the peptides were characterized by liquid chromatography ion trap time-of-flight mass spectrometry. The analytical data for the synthesized peptides are provided in Appendix A.

### 4.2. Binding Affinity to ERα

The binding assays were performed in non-binding black 384-well plates using recombinant human ERα and the fluorescent probe FAM-PERML, which is a PERM-L analogue conjugating 5(6)-carboxyfluorescein (FAM) (see Appendix A), in assay buffer to give a final volume of 10 μL. Test compounds (10 μL) prepared as stocks in DMSO were added and the plate was incubated for 2 h at room temperature. Each test compound was tested against ERα in triplicate at final compound concentrations of 10 μM, 2.5 μM, 0.63 μM, 0.16 μM, 39 nM, 9.8 nM, 2.4 nM and 0.61 nM. Plates were then read with excitation (470 nm) and emission (530 nm) wavelengths using an EnVision Multilabel Plate Reader (PerkinElmer, Waltham, MA, USA). The measurements of fluorescent polarization of a molecule (*mP*) were taken in the fluorescent polarization mode. The percent inhibition by test compounds was calculated according to the following equation:(1)$$\mathrm{Percent}\;\mathrm{inhibition}=\left(\frac{mP_{100\%}\;-\;mP_{\mathrm{sample}}}{mP_{100\%}}\right)\times 100$$
where *mP*_sample_ is the value of the wells containing test compounds and *mP*_100%_ is the value of the maximum binding well. The concentration of test compounds that reduces *mP* by 50% (IC_50_) was estimated from a graph that plotted *mP* versus the concentration of the compounds on a semi-log axis.

### 4.3. Cell Culture

Human breast carcinoma MCF-7 cells were maintained in RPMI 1640 medium containing 10% fetal bovine serum and 100 µg/mL kanamycin. The cell line was purchased from ATCC (Manassas, VA, USA), and their cell authentications were confirmed by morphology, karyotyping, and PCR-based approaches in ATCC. The database name of the cell line (MCF-7): Catalogue of Somatic Mutations in Cancer (COSMIC); the accession number: COSS905946.

### 4.4. Western Blot Analysis

MCF-7 cells were seeded in 12-well plates at a density of 1.5 × 10^6^ cells/well and cultured for 24 h. The cells were treated with the indicated peptides and incubated for indicated periods. Then, the cells were washed with phosphate buffer (PBS), lysed with SDS lysis buffer (0.1 M Tris-HCl [pH 8.0], 10% glycerol, 1% SDS) and immediately boiled for 10 min to obtained clear lysates. The protein concentration was measured by the BCA method (Pierce, Rockford, IL, USA) and the lysates containing an equal amount of protein were separated by SDS-PAGE and transferred to PVDF membranes (Millipore, Darmstadt, Germany) for western blotting analysis using the appropriate antibodies. The immunoreactive proteins were visualized by using the Immobilon Western chemiluminescent HRP substrate (Millipore) or Clarity Western ECL substrate (Bio-Rad, Hercules, CA, USA), and light emission was quantified with a LAS-3000 lumino-image analyzer with Image Gauge version 2.3 software (Fuji, Tokyo, Japan). The antibodies used in this study were anti-ERα rabbit monoclonal antibody (Cell Signaling Technology, Danvers, MA, USA; 8644) and anti-Lamin B goat polyclonal antibody (Santa Cruz, Dallas, TX, USA; sc-6216).

### 4.5. Reporter Gene Assay

ER-antagonistic activities of compounds were evaluated by a reporter assay using the Dual-Luciferase^®^ Reporter Assay system (Promega, Madison, WI, USA). MCF-7 cells were transfected with the firefly luciferase reporter plasmid containing three tandem copies of the estrogen-response element and control Renilla luciferase plasmid-SV40 using Lipofectamine LTX (Life Technologies, Inc., Gaithersburg, MD, USA). After 16 h, cells were treated with the indicated concentrations of test compounds in the presence of 17β-estradiol (3 nM). The firefly luciferase activity in cell lysates was determined based on the fluorescence intensity using a plate reader (ARVO SX1420, Perkin Elmer, Inc., Waltham, MA, USA) and normalized to the Renilla luciferase activity.

## Data Availability

The data presented in this study are available on request from the corresponding author.

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
