# Peer review of "Peptide Stapling Improves the Sustainability of a Peptide-Based Chimeric Molecule That Induces Targeted Protein Degradation"

_ijms, 2021, doi:10.3390/ijms22168772_

Round 1

Reviewer 1 Report

Manuscript No.: ijms-1322041

Peptide Stapling Improves the Sustainability of a Peptide-Based Chimeric Molecule that Induce Targeted Protein Degradation

Yokoo et al. engineered the stapled peptide stPERML-R7, which binds to the estrogen receptor alpha (ERa) and recruits E3 ligases for polyubiquitylation of ERa and subsequent proteasomal degradation. StPERML-R7 consists of the ERabinding peptide PERML, stabilized by hydrocarbon stapling (st), thereby overcoming the redox sensitivity of the previously developed PERM3 peptide. In addition, StPERML-R7 contains a cell penetrating hepta-arginine peptide (R7). The authors describe the synthesis of stPERML-R7 and test its ability to (i) interact with ERa, (ii) decrease cellular ERalevel and (iii) reduced the activity of ERa-receptor as determined in a luciferase-based reporter gene assay.

In general, the study is carefully designed and technically sound. However, the authors could improve the presentation of their results by following the suggestions listed below.

Principle criticism:

  1. In general, the figure legends are not informative and don’t contain enough information to understand the figures. Very importantly, abbreviations should be explained. For details see below (points 7, 8, 9 and 14).
  2. While the introduction provides the reader with a reasonable amount of background information and detail, the results part does not adequately describe the experimental strategies and outcomes. A few more words would be appreciated.
  3. On the other hand, the first paragraph of the results section (lines 91-107) reads like a detailed description of a method and accordingly should be shifted to Materials and Methods. However, a less technical description of the peptide synthesis and the underlying rational would be an adequate opener of the results part.

Major criticism:

  1. Figure 4: It seems, the experiment has not been reproduced. The experiment definitely should be done in duplicates or better in triplicates, as has been done in Figure 5. Quantitation of the ERa amount (e.g., as bar chart) with error bars should be performed.

Minor criticism:

  1. Line 40: why are pathogenic proteins difficult to target with conventional inhibitors?
  2. Line 85: What is the reason for stapling?
  3. Figure 1 legend: indicate that ERa stands for estrogen receptor a. Write out the abbreviations used in the figure such as: Ub, IAP, VHL, CRBN etc. Please give a short explanation what the scheme actually means.
  4. Figure 2 legend: Again, more details are required. Currently, the figure legend consists of a headline only.
  5. Scheme 1 legend: Same as for Figure 2.
  6. Line 129: Better write ERa, ER can be confused with endoplasmic reticulum.
  7. Line 131: Should better read: The peptide-mediated reduction…
  8. Figure 4: Please indicate in Figure legend and text that these are quantitative Western Blots.
  9. Do the authors observe a pattern of bands in their Western Blots, indicating ERa polyubiquitylation, especially when applying the proteasome inhibitor MG132?
  10. Figure 5: a) What does E2 stand for, 17ß estradiol? Please indicate in figure and text.
    b) What does 4OHT mean, 4-Hydroxytamoxifen? Please indicate in figure and text and indicate that this substance is used as a positive control or reference inhibitor for estrogen/ ERa mediated gene expression.
    c) Replace ∞M by µM.
    d) I suggest normalizing to the E2 stimulated sample (column 2) to have a better comparison how strong the inhibitory effect of the peptides is compared with the 4-OHT positive control.
  11. The discussion is very short. Maybe the authors find some more words to discuss their findings by comparing it with published data and state of the art approaches in the field?
  12. Shouldn´t the title read: …Chimeric Molecule that Induces …?

Reviewer 2 Report

In this paper, the authors developed the stapled peptide-based ERα degrader LCL-stPERML-R7. In previously, the authors developed a peptide-based ERα degrader (LCL-PERM3-R7) composed of the peptidomimetic estrogen receptor modulators 3(PERM3) which has a binding ability to ERα, a cell-penetrating peptide (R7) and a small molecule as E3 ligase ligand. However, LCL-PERM3-R7 is not stable in a cell environment because of the disulfide bond in PERM3 sequence. To overcome such problems, the author introduced the hydrocarbon-type staple structure. Compared to unstapled molecule, LCL-PERML-R7, LCL-stPERML-R7 showed higher degradation activity. This strategy will contribute to development of peptide-based target protein degradation inducers as therapeutic agents.

 I recommend publication of this paper after minor revision. My concern is as follows.

  1. Line 135-141; The authors mentioned that “The degradation activity of LCL-PERML-R7 almost disappeared after 24 h, whereas that of LCL-stPERML-R7 remained until at least 72 h. This result showed that LCL-stPERML-R7 maintained degradation activity for a longer period than LCL-PERML-R7, and this sustained activity may be due to increased intracellular stability of the peptide by stapling”. The meaning of stability is unclear. What is the meaning of stability, chimeric peptide itself or its helical structure?
  2. In figure 4(C), the differences of ERαdegradation at 48h between 10 μM of LCL-PERML-R7 and LCL-stPERML-R7 are clear. However, in figure 5, luciferase activities are almost same at same concentration. Please explain this reason.
  3. In figure 5, please explain E2 and 4-OHT.
